# Effect of Ultrasonic Pulses on the Functional Properties of Stickwater



**Francisco Cadena-Cadena [1], Dulce Alondra Cuevas-Acuña [2] , Felipe de Jesús Reynaga-Franco [1], Gilberto Rodríguez-Felix [1], María del Socorro Núñez-Ruiz [1], Odilia Azucena Higuera-Barraza [1], Alba Rocio Ochoa-Meza [1,\*] and Joe Luis Arias-Moscoso [1,\*]**

[1] Department of Engineering, Technological National of Mexico, Technological Institute of the Yaqui Valley, Bacum 85276, Mexico; fcadena.cadena@itvy.edu.mx (F.C.-C.); felipe.rf@vyaqui.tecnm.mx (F.d.J.R.-F.); grodriguez.perez@itvy.edu.mx (G.R.-F.); mnunez.ruiz@itvy.edu.mx (M.d.S.N.-R.); ohiguera.barraza@itvy.edu.mx (O.A.H.-B.)

[2] Department of Health Sciences, University of Sonora, Ciudad Obregon 85040, Mexico; dulce.cuevas@unison.mx

[\*] Correspondence: aochoa.meza@itvy.edu.mx (A.R.O.-M.); joe.am@vyaqui.tecnm.mx (J.L.A.-M.)

**Abstract:** Large volumes of waste are generated in the processing operations of the fishing industry. These effluents contain potentially useful proteins. However, it is necessary to concentrate them for utilization. The stickwater (SW) resulting from this operation was subjected to a protein-fractionation step, pH adjustment (acid + alkaline) and ultrasonic pulsing in order to aid in hydrolysis and evaluate its functional and nutritional properties. The protein fractions, as well as the protein hydrolysates present in the tail water, had a chemical composition of $54.85 \pm 4.21$ and $74.81 \pm 3.89$ protein (%), $0.8 \pm 0.1$ and $0.2 \pm 0.015$ fat (%), $7.21 \pm 0.67$% ash (%), respectively. The increase in low-molecular-weight peptides results in an increase in free-radical scavenging activity. However, the increase in ferric-reducing antioxidant power may be due to the HCl treatment performed by the company. An increase in the functional properties of the samples treated with ultrasonic pulses was observed. Therefore, the chemical, nutritional and functional characteristics of stickwater suggest its potential use as a food additive.

**Keywords:** stickwater; fractionation; chemical composition; antioxidant properties; functional properties



## 1. Introduction

Production of global fisheries has reached catch volumes of up to 179 million tons of fish, with 205 million tons forecast to be reached by 2030. Worldwide, only 50–60% of marine catches are used for human consumption [1], increasing waste and byproducts of fishing. In this industry, as much as 70–75% of catches can be considered waste or byproducts [2,3]. Waste may include entrails, skin, scales, shells, gills, dark meat, heads, bones, spines and effluent. These products are usually thrown into the environment without any treatment or added value [4,5].

In recent years, due to limited biological resources, the need has arisen to use and make the most of fishing resources so that a large part of the byproducts generated are destined for use in fishmeal and surimi-like products [6]. However, with the elaboration of these products, an intermediate effluent commonly called stickwater (SW) is generated [7].

SW has a high concentration of insoluble matter on its surface (5–10% solids), mainly comprising proteins, amino acids, and collagen. This high concentration of biomolecules causes the deterioration of ecosystems in which it is discharged [8,9]. That is why alternatives have been sought for its use. Among the technologies that stand out is ultrasonic pulses as pretreatment to produce protein hydrolysates. Ultrasound affects hydrogen bonds and hydrophobic interactions by alternating molecular conformation through cavitation, heating, vigorous agitation, and shear stress [10]. Ultrasound pulses create high-

and low-pressure regions that induce a series of compressions and decompressions, causing molecular displacement by interrupting dipole–dipole forces, hydrogen bonds, and other weaker interactions (high-intensity ultrasound in the range of 10–1000 W/cm$^2$ with frequencies of 20–100 kHz) [9,11].

Additionally, this technique aid in extraction of functional compounds with improved performance, short production time, low cost, and less solvent used [12]. Recent studies have focused on identifying the type of proteins present in SW. However, most have only focused on collagen/gelatin, one of the most abundant proteins in this byproduct [13], leaving aside other important protein fractions that can also possess functional and antioxidant properties. Therefore, the aim of this study aims is to produce protein hydrolysates from the soluble fraction of SW by using ultrasonic pulses, characterizing its functional properties (solubility, foaming, and emulsifying capacity) and antioxidant properties (ABTS, DPPH, and FRAP).

## 2. Materials and Methods

### 2.1. Raw Material

Stickwater was provided by Pescaharina de Guaymas S.A. de C.V., located in Guaymas, Sonora, Mexico (27°55′09″ N). SW from a batch containing the species California anchovy (*Engraulis mordax*) and Pacific anchovy (*Cetengraulis mysticetus*) was collected directly from storage tanks in high-density polyethylene gallons and quickly placed on ice, where it was stored at −20 °C until further analysis.

### 2.2. Proximal Chemical Analysis of Stickwater

Proximal chemical analysis of SW was performed according to the methods of the AOAC (Association of Official Analytical Chemists, Rockville, MD, USA, 1990): 950.46, 925.23, 920.153, 981.1, and 991.36 for moisture, solids, ashes, protein, and fat, respectively. The results are expressed as a percentage.

### 2.3. Soluble Stickwater Protein

Soluble protein content was measured following the methodology described by Bradford [14]. Bovine serum albumin was used as a standard (1 mg/mL), and absorbance was quantified at 595 nm in a spectrophotometer (Cary 50; Varian; Palo Alto, CA, USA).

### 2.4. Proteins Recovery from Stickwater

The stickwater protein-separation process is shown in Figure 1. Protein fractionation consisted of mixing 70 g stickwater with 40 mL of a solution to extract a particular protein fraction. Sarcoplasmic protein was extracted from SW with pH 7.5 0.2 M NaCl phosphate buffer solution. Myofibrillar fraction solubilization was performed with a pH 7.5 2 M NaCl phosphate buffer. Finally, the stromal fraction was obtained with a 2 M NaOH solution. Each of the samples was homogenized for 30 s. Then, a centrifuge was used to keep the soluble fraction at 13,000× $g$ for 15 min at 20 °C (Thermo Scientific; Sorvall Biofuge Stratos; Waltham, MA, USA). All supernatants were additionally filtered with 125 mm Whatman paper to remove insoluble material residues [15]. Recovered protein fractions were stored under refrigeration for a period no longer than 15 days [15,16]. Ultrasound pulses were applied under the following conditions: 30% amplitude for 10 min, with a pulse on-and-off time of 10 s, using an ultrasonic processor (VCX750, Vibra cell, Sonics; Newtown, CT, USA) [17]. A total of 70 g of SW was placed in beakers surrounded by an ice bath to avoid increasing temperatures. Immediately afterward, protein concentration [14] and antioxidant activity of SW were determined [18,19].

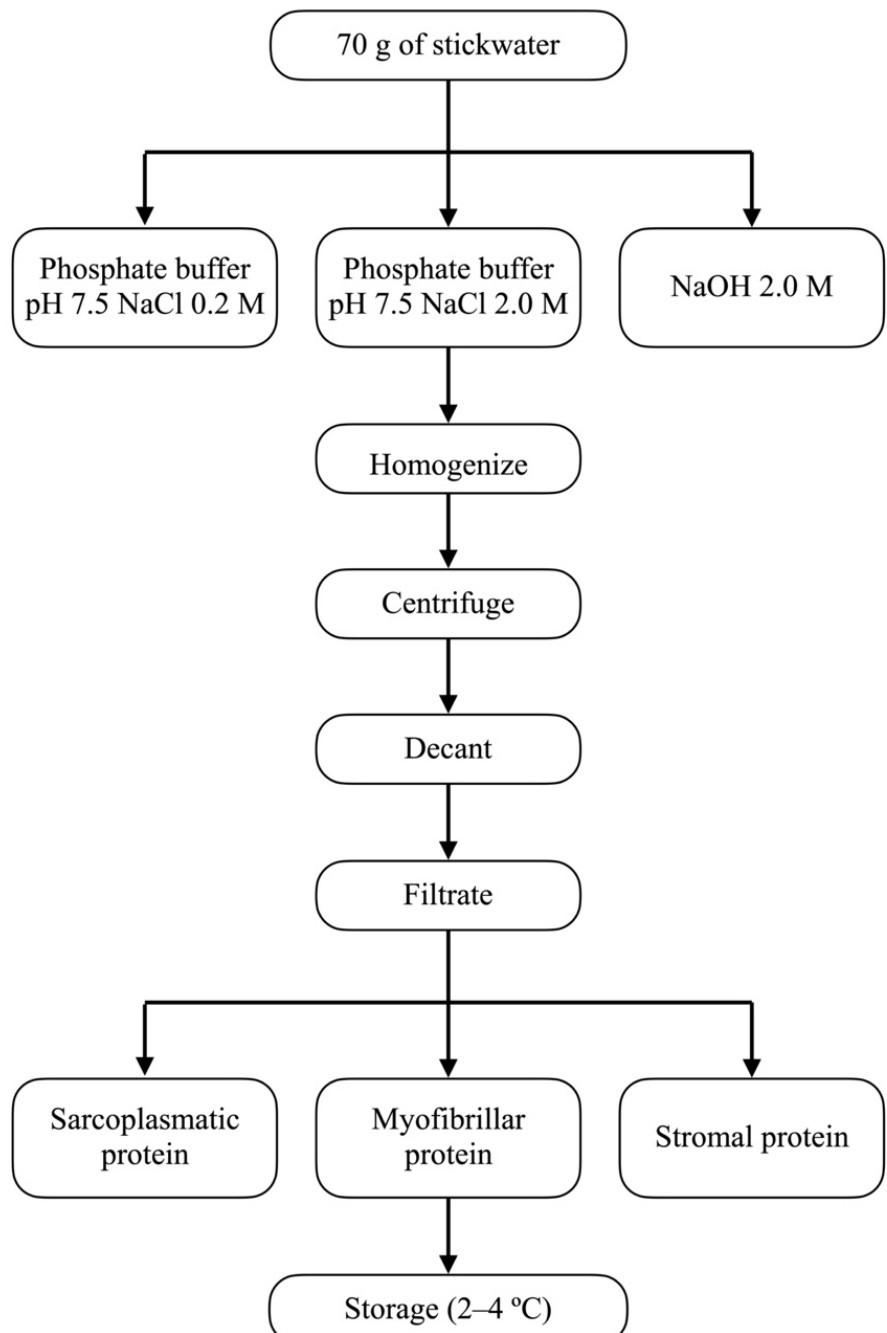

**Figure 1.** General diagram of the stickwater protein-separation process.

*2.5. Proximal Composition of Stickwater*

The proximal composition of the SW and the protein fractions extracted through ultrasonic pulses (humidity, ash, fat, and proteins—the latter reported as the percentage of nitrogen multiplied by a factor of 6.25) were determined using the official methodologies implemented by the Association of Analytical Chemists [20].

*2.6. Functional Properties of Protein Fractions of Stickwater*

2.6.1. Emulsifying Stability (ES)

Emulsifying stability was evaluated using the modified methods of Sathivel et al. [9]. The emulsion was prepared by homogenizing the sample (500 mg) for 2 min at 1800 g with 50 mL of 0.1 M NaCl and gradually adding 50 mL of soybean oil (Purela, Escuintla, Guatemala). Aliquots of 25 mL were taken and transferred to a 25 mL test tube. The

emulsions were allowed to stand for 15 min at 25 °C; then, the separated aqueous volume was read. ES (%) was calculated as follows:

$$ES\ (\%) = ((total\ aqueous\ volume\ separated\ by\ volume)/(total\ volume)) \times 100$$

### 2.6.2. Foaming Capacity (FC)

Foaming ability was evaluated by the methods modified by Wild and Clark [21], with some additional modifications. A total of 60 mg of each stickwater fraction (dry matter) was solubilized and separated in 30 mL of Tris-HCl buffer solution, pH 6.8, to obtain a final concentration of 2 mg/mL. Portions of 20 mL of protein solution were homogenized at 1800 g for 1 min using a tissue homogenizer (Wisd, WiseTisHG-15D, Witeg, Germany). The solutions were transferred to a 25 mL test tube [21]. The foam volume at 30 s was calculated as follows:

$$FC\ (\%) = ((Foam\ volume\ (mL))/(initial\ liquid\ volume\ (mL))) \times 100$$

### 2.6.3. Antioxidant Properties of the Protein Fractions of Stickwater

The antioxidant capacity of the protein fractions recovered from the stickwater was determined by means of 2,2'-Azinobis-3-ethyl-benzothiazoline-6-sulfonic acid (ABTS), and the antioxidant-reducing power of ferric ions (FRAP) was determined using a spectrophotometer (Cary 50; Varian; Palo Alto, CA, USA).

ABTS radical-scavenging activity was carried out according to the method described by Re et al. [18]. Portions of 50 mg of each stickwater fraction were solubilized to separate them in 1 mL of Tris-HCl buffer solution at pH 6.8. The final concentration of the dilutions was 25, 12.5, 6.25, 3.12, 1.56, and 0.78 mg/mL. The ABTS radical was prepared with 88 µL of $K_2S_2O_4$ (0.14 mM) in 5 mL of ABTS + (7 mM) and incubated for 16 h in the dark at room temperature. A total of 270 µL of ABTS solution was mixed with 20 µL of each protein solution to separate, taking 30 min to read at 734 nm. The results were calculated as percent inhibition of the radical ABTS.

The reducing antioxidant power of ferric ions (FRAP) was determined according to the method described by Benzie and Strain [19], with some modifications. This method is based on the ability to quickly reduce $Fe^{3+}$ ions to $Fe^{2+}$ in the presence of 2,4,6-Tripyridyl-s-triazine (TPTZ). A total of 20 µL of the sample was added to 280 µL of FRAP solution and incubated for 30 min in a darkroom. Absorbance was measured at 630 nm in a microplate reader, and the results were calculated as percent inhibition of the radical FRAP.

### 2.7. Electrophoretic Profile

Sodium dodecyl sulfate-polyacrylamide gel electrophoresis (SDS-PAGE) was performed for possible determination and identification of proteins in the sample. The gels were prepared discontinuously (gel less than 12% and gel greater than 4%, respectively), with a thickness of 1 mm. The samples were mixed with SDS-PAGE sample buffer (4% SDS, 20% glycerol, 10% b-mercaptoethanol, 0.125 M Tris, pH 6.8) and dissolved by heating in boiling water for 3 min. Electrophoresis assays were carried out on a Bio-Rad Power Pao 3000 (Bio-Rad Lab., Hercules, CA, USA) using 25 mA per gel. Proteins were visualized with Coomassie blue staining [22]. The molecular weight of the protein was estimated using wide-range standards [23,24].

### 2.8. Degree of Hydrolysis

The degree of hydrolysis (DH) was analyzed using the o-phthalaldehyde (OPA) method [25]. The OPA method is based on a reaction between amino groups and OPA in the presence of a thiol group, from which a colored compound is formed, which is detectable at 340 nm. The milliequivalents (meq) of released *Ser-NH₂* were calculated as follows:

$$Ser - NH_2 = \left[ \frac{Abs\ sample - Abs\ blank}{Abs\ standard - Abs\ blank} \right] \left( 0.96516 \frac{mEq}{L} \right) (0.1) \left( \frac{100}{XP} \right)$$

where *Ser-NH$_2$* = (*mEq·Ser-NH$_2$*)/*g* protein; *X* is the mass (*g*) of the reaction mixture; *P* is the protein percentage in the sample; and 0.1 is the sample volume in liters (*L*) [25].

Next, hydrolysis (*h*) was calculated as follows:

$$H = \frac{Ser - NH_2}{\alpha\ megv/g\ protein}$$

where *β* and *α* applied to stickwater are 0.40 and 1.0, respectively [26,27].

Lastly, the *DH* was calculated as follows:

$$DH = (h/htot) * 100$$

where *htot* used for the stickwater is 8.6 [26,27]. *DH* is expressed as the total percentage of hydrolyzed protein.

### 2.9. Statistical Analysis

Proximal composition data were analyzed using an analysis of variance (ANOVA), expressed as the value ± the standard deviation of four repetitions, while the results obtained from the electrophoretic profile were analyzed using descriptive statistics.

## 3. Results and Discussion

### 3.1. Proximal Composition of Stickwater

The proximal chemical composition of stickwater is shown in (Table 1). In the present study, 9% of solids, 0.8% of fat, 54.85% of protein, 53.84% of moisture, and 7.21% of ash (dry basis) were obtained in the treatment without ultrasonic pulses.

**Table 1.** Proximal chemical composition of Stickwater.

| Stickwater | Moisture | Solids | Ashes | Protein | Fat |
| --- | --- | --- | --- | --- | --- |
| Without pulsation | 53.8 ± 3.6 | 9.0 ± 0.8 | 7.2 ± 0.6 | 54.8 ± 4.2 | 0.8 ± 0.1 |
| Ultrasound-pulsed | 66.0 ± 2.3 | 6.4 ± 0.3 | 7.2 ± 0.6 | 74.8 ± 3.8 | 0.2 ± 0.0 |

Results are expressed as the average of five determinations.

Submitting stickwater to ultrasound pulses modified protein and solid values (74.81 and 6.45, respectively). Goycoolea (1997) reported 8–10% of total solids, 5.6% protein, 0.6% fat, 1.8% ash, and 92% humidity in SW from a flour industry in Ensenada, Baja California. On the other hand, in SW, from a mixture of *Sardinops sagax caerulea*, *Engraulis mordax*, *Cetengraulis mysticetus*, and *Scomber japonicus*, 9.5% total solids, 4.7% protein, 1.7% ash, and 1.8% lipids were obtained [28]. Valdéz [29] found lower amounts of protein in SW of Monterrey sardine (*Sardinops sagax cerulea*), with 91%, 3%, 5.1%, and 1.4% for moisture, protein, fat, and ash, respectively. According to various authors, protein content may vary depending on capture time, size of the individuals, and treatment after the recovery of the effluent. Additionally, the extraction methodology and the protein-measurement technique (dry or wet basis) can lead to variations in the measurement of proteins [7,30]. Through centrifugation and acid precipitation with TCA and HCl, other authors separated the insoluble-protein fraction of stickwater from the species above, obtaining its chemical composition on a dry basis. The fraction treated with HCl presented contents of 69.7%, 8.2%, 13.9%, and 4% for protein, ash, fat, and non-protein nitrogen, respectively. In contrast, for the fraction treated with TCA, the reported values were 76.7%, 7.9%, 12.1%, and 3.2% in protein, ash, fat, and non-protein nitrogen, respectively [31].

The significant variation in the proximal content in this study resulted because the stickwater was subjected to an industrial evaporation process before the sample was obtained, which considerably reduced the amount of moisture. At the same time, components such as protein and ash were concentrated. It should be mentioned that said elimination of water was not enough for the sample to be taken on a dry basis. However, the proximal

composition of SW depends on the species of fish, the time of capture, and physiological conditions [7].

### 3.2. Identification of Stickwater Proteins

The molecular weight of the proteins present in the stickwater was determined using a polyacrylamide SDS gel. The molecules present in the gel had molecular weights ranging from 6 to 200 kDa (Figure 2). Molecular weights of 200 kDa suggest the presence of the heavy chain of myosin, especially in those solutions treated with concentrations of 0.2 M and 2 M NaCl. These results are expected due to the concentration of muscle proteins in the effluent. During the fishmeal manufacturing process, one of the steps consists of pressing the fish, and an essential part of the muscle tissue is drawn into the pressing effluents [32]. Other proteins found in Alaska pollock SW and cod are troponin or tropomyosin, with a molecular weight between 32 and 34 kDa. However, in other studies, the proteins found with the highest concentration had molecular weights of 39, 60, 120, and 198 kDa, consistent with those found in this study [7].

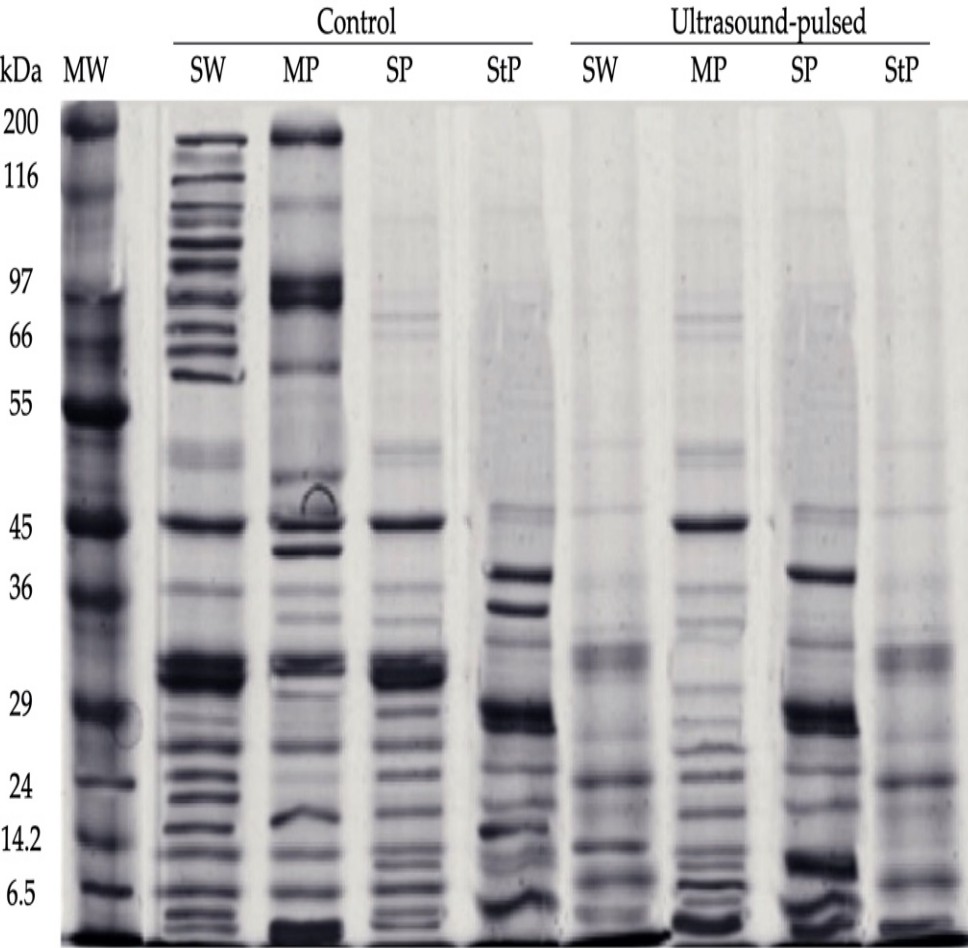

**Figure 2.** Protein patterns of anchoveta stickwater with and without ultrasonic pulses. MW, molecular weight standard; SW, stickwater; MP, myofibrillar protein; SP, sarcoplasmic protein; StP, stromal protein.

Finally, the presence of low-molecular-weight peptides (6 kDa) is considered a product of protein degradation and denaturation during thermal processing and pressing of fish [7,30]. However, more research is needed to determine the origin of these molecules.

### 3.3. Emulsifying Capacity (EC) and Foaming Capacity (FC)

Proteins are of utmost importance to food science, as they are used as stabilizers in food emulsions; this is of great concern because many foods are emulsions. [33]. The results of the EC and FC studies of SW and its treatment with ultrasonic pulses are shown in Table 2.

**Table 2.** Functional properties of stickwater (SW).

|  |  | Emulsifying Capacity (%) | Foaming Capacity (%) | ABTS (%) | FRAP (%) |
|---|---|---|---|---|---|
| Stickwater | C | 65.4 ± 3.2 [b] | 62.9 ± 5 [a] | 45.2 ± 2.2 [b] | 70.4 ± 2.2 [c] |
|  | UP | 88.3 ± 5.8 [c] | 98.4 ± 2 [c] | 73.8 ± 4.3 [c] | 90.5 ± 4.0 [d] |
| Sarcoplasmic protein | C | 44.7 ± 4.8 [a] | 85.8 ± 1 [b] | 39.1 ± 3.5 [ab] | 24.8 ± 2.5 [b] |
|  | UP | 86.8 ± 4.1 [c] | 98.3 ± 4 [c] | 83.7 ± 4.4 [c] | 91.8 ± 3.2 [d] |
| Myofibrillar protein | C | 35.2 ± 2.8 [a] | 69.7 ± 5 [a] | 32.8 ± 3.6 [a] | 14.4 ± 2.1 [a] |
|  | UP | 76.6 ± 7.8 [bc] | 99.5 ± 3 [c] | 73.8 ± 5.6 [c] | 97.7 ± 3.3 [d] |
| Stromal protein | C | 34.1 ± 5.2 [a] | 67.3 ± 3 [a] | 46.7 ± 1.7 [b] | 33.5 ± 4.5 [b] |
|  | UP | 87.5 ± 2.8 [c] | 98.2 ± 2 [c] | 98.2 ± 1.8 [d] | 98.6 ± 0.9 [d] |

The results are expressed as the average of four determinations. Control (C), Ultrasound pulsed (UP). [a,b,c,d] Different letters indicate significant differences between rows.

The presence of ultrasonic pulses increased functional properties by 20 to 45%. This increase could be due to the denaturation suffered by the protein due to the ultrasonic pulses. Protein hydrolysates form hydrophilic and hydrophobic peptides, which is why they can be considered surfactant materials that help form oil–water emulsions. Emulsifying capacity is related to the size, type of protein, and type of peptide that is interacting at the interface [34]. When the formed emulsion is weak, it is due to the production of hydrophilic peptides that are located in the aqueous phase [35]. Therefore, protein concentration is related to the stability of the emulsions, since at low protein concentrations, the oil–water interface is controlled by diffusion. In contrast, at high protein concentrations, the activation energy does not allow for the migration of proteins in a diffusion-dependent manner, leading to aggregation of long-chain proteins and peptides [16,36].

The increase in foaming properties of proteins treated with ultrasonic pulses is related to solubility, pH, and the type of ion present in the protein solution [37,38]. In sarcoplasmic proteins from rohu (*Labeo rohita*) [36], cod (*Gadus morhua*) [39], and sea bream (*Nemipterus hexodon*) an increase in foaming capacity has been found to be derived from an increase in pH on the primary side and the concentration of NaCl [38]. The sum of acid residues in fish proteins is greater than the sum of basic residues; therefore, minimum solubility is observed at an approximate pH of 4–5, and maximum solubility at alkaline pH [40,41]. In this sense, the further a protein is from its isoelectric point, the better its foaming capacity. Similar results were reported for the devilfish and for giant squid [16,41]. Extraction of proteins was carried out with different concentrations of NaCl, which directly impacts the formation of foam; when treating proteins with ultrasonic pulses, solubility increases due to denaturation. However, high NaCl concentrations decrease EC formation, possibly caused by aggregation and increased exposure to hydrophobic groups [42,43].

### 3.4. Antioxidant Capacity and Degree of Hydrolysis

In vitro ABTS and FRAP assays determined antioxidant capacity. ABTS assay can determine hydrophilic and lipophilic hydrogenated radicals [44]. The low antioxidant capacity of the effluents before ultrasonic pulses is related to their low solubility in water and the low availability of antioxidant molecules. However, after applying the ultrasonic pulses, an increase of more than 50% in antioxidant activity against the radical ABTS can be observed, possibly due to protein cleavage, the availability of low-molecular-weight peptides, and the presence of antioxidant AAs (such as Ala and Phe) [45]. An excessive degree of hydrolysis and a poor degree of hydrolysis cause incomplete removal of the

ABTS radical. The degree of hydrolysis of the samples when using ultrasonic pulses was 10%, which directly influenced the size of the peptides present in the hydrolysate. Similar results were found in the SW of *Clupeonella* sp. [46] and Pacific thread herring (*Ophistonema Libertate*) [47]. High activity of ABTS in fish wastewater has been associated with hydrophobic amino acids (Val, Ala, and Leu) at the amino terminus position [42]. With their amino-acid composition, peptide sequences such as Leu-Leu-Pro -His-His, His-Pro-Val, or Pro-His-His are considered antioxidants. Likewise, peptides that contain His in their structure can act as metal chelators and reactive-oxygen-species scavengers [47,48].

Treatment of the sample with HCl performed by the company allows for maintenance of the iron-reducing activity (FRAP) (it indicates whether a compound can donate electrons to a free radical to neutralize it) of the wastewater before treatment with ultrasonic pulses. However, antioxidant activity was increased when ultrasonic pulses were used, compared to raw sewage. Similar results have been obtained for the wastewater of Pacific thread herring (*Ophistonema Libertate*) [49], tuna (*Thunnus tonggol*) [50], cuttlefish (*Sepiida*) [51], and shrimp (*Penaeus* spp.) [52].

The increase in the FRAP iron potential is related to the presence of chelating amino acids in the peptic sequence (Cis, His, Glu); the SH groups interact directly with free radicals. In addition, amino acids, such as Ala, Leu, and Gly enhance this activity when they are found in the peptic sequences. The hydrogen donor and peroxyl-radical-trapping capacities are attributable to the imidazole group as part of the His residue, making it one of the best antioxidant amino acids. Additionally, adding leucine or proline to the N terminal of a histidine peptide improves antioxidant activity. Therefore, the position in the peptide sequence plays an essential role in the antioxidant activity of the different proteins from wastewater resulting from the manufacture of anchovy fishmeal by FRAP [48,51].

### 4. Conclusions

This study confirms the presence of functional properties in anchovy stickwater. It is necessary to emphasize the continuation of studies to characterize and generate more information on the proteins present in stickwater with a wide potential for use in the pharmaceutical or food industries. Furthermore, the information generated on these proteins may form the basis for future studies of the recovery of proteins with functional properties, antioxidant properties, and nutraceutical products from effluents of the processing of the fishing industry. The use of effluents such as SW can help to minimize the environmental impact of these effluents on coastal areas while also increasing their added value.

**Author Contributions:** Conceptualization, F.C.-C. and J.L.A.-M.; methodology, O.A.H.-B.; software, G.R.-F.; validation, F.d.J.R.-F.; formal analysis, A.R.O.-M.; investigation, O.A.H.-B. and D.A.C.-A.; resources, J.L.A.-M.; data curation, M.d.S.N.-R.; writing—original draft preparation, F.C.-C.; writing—review and editing, D.A.C.-A. and F.C.-C.; visualization, F.d.J.R.-F.; supervision, J.L.A.-M.; project administration, A.R.O.-M. and G.R.-F.; funding acquisition, F.C.-C. All authors have read and agreed to the published version of the manuscript.

**Funding:** This research received no external funding.

**Institutional Review Board Statement:** Not applicable.

**Acknowledgments:** The authors acknowledge the support of the University of Sonora through lab use and the Technological Institute of the Yaqui Valley.

**Conflicts of Interest:** The authors declare no conflict of interest.

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
