# Peer review of "Effect of Ultrasonic Pulses on the Functional Properties of Stickwater"

_applsci, doi:10.3390/app12031351_

Round 1
Reviewer 1 Report
The submitted manuscript tends to describe the analysis of stickwater, after the effect of ultrasound pulses. The work is interesting and fits within the scope of the Special Issue. The analysis of stickwater consisted of the determination of moisture, solids, ashes, protein, and fat using official methods. The functional properties (solubility, foaming, and emulsifying capacity) and antioxidant properties (ABTS, DPPH, and FRAP) of protein fractions of stickwater was also examined, using known methods. The results are seem to be precise and the authors discuss them in depth. Revisions have to be undertaken by the authors according to the following comments:
Specific comments
- Line 20: please correct the numbers “12 ± 0.8± 0.1”
- The Instrumentation section is missing.
- Please revise “ml” as “mL”.
- Lines 78-84 are not supported by reference, neither a discussion about the use of phosphate buffer, NaCl and NaOH is referred.
- Lines 80-81: “For sarcoplasmic protein, a phosphates buffer pH 7.5 0.2 M NaCl. For myofibrillar protein, a phosphate buffer pH 7.5 with 2 M NaCl. For stromal fraction, a 2 M NaOH solution.” These sentences do not make sense (verbs are missing).
- Line 85: “Later they were recovered and stored under refrigeration for a period no longer than 15 days.” However, stability studies or citation are missing.
- The ultrasound pulses, which were applied, was 30% amplitude for 10 minutes, with a pulse on and off time of 10 seconds. Why? These conditions are supported by the reference 15 (Line 87), but this paper does not mention them.
- Revise the “technique” as “methods” in Lines 105 and 113. They are methods, not techniques.
- Line 126-127: 50 mg is solubilized in 1 mL of buffer. Why the final concentrations are referred as g/mL in the next sentence?
- Table 1: The caption of the table is missing. Furthermore, what does "control" mean? There is no discussion about it.
Author Response
Response to Reviewer 1 Comments
Specific comments
Point 1: Line 20: please correct the numbers “12 ± 0.8± 0.1”
Response 1: The authors acknowledge the miss-wording. This section has since been modified, and changes are highlighted in blue.
Point 2: The Instrumentation section is missing.
Response 2: Successful observation. The writing of the paper was checked, and modifications were made and highlighted in blue.
Point 3: Please revise “ml” as “mL”.
Response 3: The change was made successfully.
Point 4: Lines 78-84 are not supported by reference, neither a discussion about the use of phosphate buffer, NaCl and NaOH is referred.
Response 4: The change was made successfully.
Point 5: Lines 80-81: “For sarcoplasmic protein, a phosphates buffer pH 7.5 0.2 M NaCl. For myofibrillar protein, a phosphate buffer pH 7.5 with 2 M NaCl. For stromal fraction, a 2 M NaOH solution.” These sentences do not make sense (verbs are missing).
Response 5: Successful observation. The writing of the paper was checked, and modifications were made and highlighted in blue.
Point 6: Line 85: “Later they were recovered and stored under refrigeration for a period no longer than 15 days.” However, stability studies or citation are missing.
Response 6: The change was made successfully.
Point 7: The ultrasound pulses, which were applied, was 30% amplitude for 10 minutes, with a pulse on and off time of 10 seconds. Why? These conditions are supported by the reference 15 (Line 87), but this paper does not mention them.
Response 7: The change was made successfully and changes are highlighted in blue
Point 8: Revise the “technique” as “methods” in Lines 105 and 113. They are methods, not techniques.
Response 8: The change was made successfully.
Point 9: Line 126-127: 50 mg is solubilized in 1 mL of buffer. Why the final concentrations are referred as g/mL in the next sentence?
Response 9: The change was made successfully
Point 10: Table 1: The caption of the table is missing. Furthermore, what does "control" mean? There is no discussion about it.
Response 10: The change was made successfully

Reviewer 2 Report
Coments for the manuscript Applied science-1546901
Articles that try to take advantage of the huge amounts of waste from manufacturing companies are always interesting.
From my point of view, only some editing errors should be corrected and some things that heve not been clear to me should be clarified
In line 19-20, review the data, the number of decimals places are not homogeneous, a data is missing from ash and what is 12 ± 0.8 ± 0.1?
In line 64 you must write “SW” instead of “Sw”.
In point 2.4 Proteins recovery from stickwater, line 77-89, a decantation is carried out after a energetic centrifugation at 13000 rpm. I understand that a centrifugation is more effective in removing solids than a decantation. Can you explain the need for this decantation?
In line 157 it is not clear that a sample can heve a solids content of 9% and a moisture content of 53.84%, what happens with the meaning 37.16%. The solids to which you refer are not the residue of the sample when the moisture is removed?. On the other hand, the protein, fat and ash content should be on dry basis, specify if this is the case or clarify these data better for a better understanding.
In line 183, what is the title of table 1?
In line 253 say, the degree of hydrolysis of the samples when using the ultrasonic pulses was 10%. With what method has the degree of hydrolysis been determined?, include it in the materials and methods section
In line 254 you must write “hydrolysate” instead of “hydrolysate”

Author Response
Response to Reviewer 2 Comments
Point 1: In line 19-20, review the data, the number of decimals places are not homogeneous, a data is missing from ash and what is 12 ± 0.8 ± 0.1?
Response 1: The authors acknowledge the miss-wording. This section has since been modified, and changes are highlighted in green.
Point 2: In line 64 you must write “SW” instead of “Sw”.
Response 2: Successful observation. The writing of the paper was checked, and modifications were made.
Point 3: In point 2.4 Proteins recovery from stickwater, line 77-89, a decantation is carried out after a energetic centrifugation at 13000 rpm. I understand that a centrifugation is more effective in removing solids than a decantation. Can you explain the need for this decantation?
Response 3: The reviewer is thanked for the comment. The writing of the paper was checked, and some modifications were made to clarify the ideas, all changes are highlighted in green throughout the text
Point 4: In line 157 it is not clear that a sample can heve a solids content of 9% and a moisture content of 53.84%, what happens with the meaning 37.16%. The solids to which you refer are not the residue of the sample when the moisture is removed?. On the other hand, the protein, fat and ash content should be on dry basis, specify if this is the case or clarify these data better for a better understanding.
Response 4: The writing of the paper was checked, and some modifications were made to clarify the ideas, all changes are highlighted in green throughout the text
Point 5: In line 183, what is the title of table 1?
Response 5: Successful observation. The writing of the paper was checked, and modifications were made.
Point 6: In line 253 say, the degree of hydrolysis of the samples when using the ultrasonic pulses was 10%. With what method has the degree of hydrolysis been determined?, include it in the materials and methods section
Response 6: Successful observation. The writing of the paper was checked, and modifications were made, all changes are highlighted in green
Point 7: In line 254 you must write “hydrolysate” instead of “hydrolysate”
Response 7: The change was made successfully.

Round 2
Reviewer 1 Report
The authors prepared a fair revision of the original submission.
In my opinion, the revised version can be accepted in its current form.